# Rare Neuronal, Glial and Glioneuronal Tumours in Adults

**DOI:** 10.3390/cancers15041120

**Published:** 2023-02-09

**Authors:** Nicolas Crainic, Julia Furtner, Johan Pallud, Franck Bielle, Giuseppe Lombardi, Roberta Rudà, Ahmed Idbaih

**Affiliations:** 1Sorbonne Université, Institut du Cerveau—Paris Brain Institute—ICM, Inserm, CNRS, AP-HP, Hôpital Universitaire La Pitié Salpêtrière, DMU Neurosciences, Service de Neurologie 2, 75013 Paris, France; 2Department of Neurology, University Hospital of Brest, 29200 Brest, France; 3Department of Biomedical Imaging and Image-guided Therapy, Medical University of Vienna, 1090 Vienna, Austria; 4Research Center of Medical Image Analysis and Artificial Intelligence (MIAAI), Danube Private University, 3500 Krems, Austria; 5Service de Neurochirurgie, GHU Paris Psychiatrie et Neurosciences, Hôpital Sainte-Anne, 75014 Paris, France; 6Institute of Psychiatry and Neuroscience of Paris, IMABRAIN, INSERM U1266, Université de Paris, 75014 Paris, France; 7Sorbonne Université, Institut du Cerveau—Paris Brain Institute—ICM, Inserm, CNRS, AP-HP, Hôpital Universitaire La Pitié Salpêtrière, DMU Neurosciences, Service de Neuropathologie, 75013 Paris, France; 8Department of Oncology, Oncology 1, Veneto Institute of Oncology-IRCCS, 35128 Padua, Italy; 9Division of Neurology, Castelfranco Veneto and Treviso Hospitals, 31033 Treviso, Italy; 10Department of Neuro-Oncology, University of Turin, 10126 Turin, Italy

**Keywords:** glioneuronal tumours, neuronal tumours, methylation class, low-grade gliomas, seizures, MAPK, BRAF

## Abstract

**Simple Summary:**

Glioneuronal and neuronal tumours are rare and mostly found in young adults and children, representing less than 5% of primary central nervous system (CNS) tumours. Accurate diagnosis is often difficult, requiring a significant body of evidence (clinical, radiological, pathology and molecular). The aim of this paper is to describe the main entities reported in the 2021 World Health Organization (WHO) classification, including, on the one hand, their histomolecular and imaging features and, on the other hand, their therapeutic management. Gross total resection is the cornerstone of the treatment of these tumours when achievable. MAPK pathway abnormalities could represent an interesting target for novel drugs.

**Abstract:**

Rare glial, neuronal and glioneuronal tumours in adults form a heterogeneous group of rare, primary central nervous system tumours. These tumours, with a glial and/or neuronal component, are challenging in terms of diagnosis and therapeutic management. The novel classification of primary brain tumours published by the WHO in 2021 has significantly improved the diagnostic criteria of these entities. Indeed, diagnostic criteria are nowadays multimodal, including histological, immunohistochemical and molecular (i.e., genetic and methylomic). These integrated parameters have allowed the specification of already known tumours but also the identification of novel tumours for a better diagnosis.

## 1. Introduction

Rare neuronal, glial and glioneuronal tumours account for less than 2% of all primary central nervous system (CNS) tumours. Histologically, these neuronal, glial, and glioneuronal tumours include in different proportions two tumour cell populations: (i) glial and/or (ii) neuronal. Some tumours are purely or predominantly neuronal (i.e., gangliocytoma, multinodular and vacuolating neuronal tumour (MVNT), neurocytoma, cerebellar liponeurocytoma), others are glioneuronal with both tumour cell populations (i.e., ganglioglioma, dysembryoplastic neuroepithelial tumour (DNET), diffuse glioneuronal tumour with oligodendroglioma-like features and nuclear clusters (DGONC), papillary glioneuronal tumour (PGT_, rosette-forming glioneuronal tumour (RGNT), diffuse leptomeningeal glioneuronal tumour (DLGNT), and others are predominantly glial (i.e., pilocytic astrocytoma, subependymal giant cell astrocytoma (SEGA), pleomorphic xanthoastrocytoma (PXA), high-grade astrocytoma with piloid features (HGAP)).

In contrast to previous versions, the new World Health Organization (WHO) classification of primary CNS tumours, published in 2021, is based on histological features and/or molecular characteristics, such as specific genetic alterations and DNA methylation classes. Indeed, some tumour types need methylation analysis for accurate diagnosis (e.g., HGAP, DGONC) [1].

All the tumour types detailed in the review (except dysplastic gangliocytoma) have their own methylation class. Nevertheless, the methylome classifier may identify the group of “low-grade glial/glioneuronal tumours” with high significance but not significantly a methylation class corresponding to a tumour type. This difficulty of methylome classification mirrors the continuum of histological aspects between tumours of this family. A pragmatic approach to inform therapeutic decision is then to consider if the tumour is circumscribed and has a druggable mutation of the MAPK pathway. Grading may be challenging in such context as criteria of high grade in one tumour type may not have any prognostic value in another tumour type. An integration of histomolecular findings with clinical and radiological courses is thus highly valuable.

The most common neurological symptoms revealing these tumour types are seizures; headaches are more rarely seen given their progressive natural history [2]. Hydrocephalus may reveal intraventricular tumours. A rare but particular symptom may be psychiatric manifestations, especially in younger patients [3]; ictal panic (misdiagnosed initially as panic attacks) has been described as the main symptom in a cohort of 10 young adults with low grade gliomas, including glioneuronal tumours [4].

The first-line treatment is surgery, preferably maximal safe gross total resection when achievable [5,6]. The place of oncological treatments, including radiotherapy and chemotherapy, in the therapeutic arsenal, is not perfectly established since the scarcity of these tumours is a limitation for phase-3 clinical trials.

Molecular targeted therapies, including MAPK [6,7,8] and NTRK signaling pathways inhibitors [9] have shown promising results.

In the current review, we will successively present: (i) neuronal tumours, (ii) glioneuronal tumours and (iii) glial tumours.

## 2. Neuronal Tumours

### 2.1. Dysplastic Cerebellar Gangliocytoma (Lhermitte–Duclos Disease, DCG)

DCG is a grade-1 gangliocytoma restricted most often to one cerebellar hemisphere and occurring in the setting of Cowden disease in one-third of cases. Cowden disease is a rare, autosomic dominant condition involving the PTEN gene. Patients present multiple and diffuse benign lesions called hamartomas, mainly on the skin, breast, and thyroid, as well as an increased risk of developing certain malignant lesions (breast, thyroid and digestive tract) [10].

Adult DCGs are mainly diagnosed between 20 and 40 years and are usually associated with *PTEN* mutations activating the PI3K-AKT-mTOR signaling pathway [11]. Small DCGs are usually asymptomatic. Over time, growing lesions can manifest with multiple symptoms, including cranial nerve palsies, cerebellar syndrome, and/or obstructive hydrocephalus.

MRI is usually sufficient for diagnosis, especially when a cerebellar folia hypertrophy with tigroid appearance (i.e., alternating hypointense and hyperintense signals on T2-weighted images) is present. There is also partial contrast uptake in half of the cases, which has been shown to be associated with abnormal venous proliferation. Those small veins running between the thickened cerebellar folia and the adjacent draining veins are both best seen on susceptibility-weighted images (SWI). The apparent diffusion coefficient (ADC) is usually elevated compared with the normal cerebellar signal; thus, a hyperintense signal on diffusion-weighted images (DWI) should not be misinterpreted as diffusion restriction but can be explained by a “T2 shine-through” effect [12]. Differential diagnosis on an MRI is usually medulloblastoma or cerebellitis [13]. Neuropathological examination shows a conserved folia architecture but with the inverted distribution of white matter and grey matter: synaptophysin-immunopositive dysplastic ganglionic cells replace granule cells, and myelinated axons replace the molecular layer. Mitoses are very rare.

If the lesion is small enough and asymptomatic, the wait-and-see approach is recommended. When it becomes symptomatic, complete surgical resection is the first-line treatment [11]. Chemotherapy and radiotherapy are not commonly used; however, a case report showed significant clinical improvement in an infant with initial bilateral cerebellum due to DCG after Rapamycin (mTOR inhibitor) initiation [14]. The overall survival rate is excellent.

### 2.2. Central Neurocytoma

Central neurocytoma (Figure 1) is a WHO grade-2 tumour occurring most commonly in the lateral and third ventricles. Most patients are between 20 and 50 years, with a male-to-female ratio close to 1. The usual clinical presentation is hydrocephalus and headaches. Psychiatric manifestations, such as psychosis and hallucinations, are quite rare but have been reported [15,16]. Incidental findings of small tumours are not rare because of their initial indolent clinical course [17].

Central neurocytomas are usually located in the lateral ventricles in the proximity of the foramen of Monro and are attached to the septum pellucidum. Accompanying ventricular dilatation is often present. On an MRI, the lesion shows a slightly hyperintense signal on T2-weighted and on fluid-attenuated inversion recovery (Flair) images with a bubbly appearance due to the cystic components and a moderate contrast enhancement [18]. Prominent flow voids may be recognized. The cysts on the tumour periphery as well as the wavy walls of the enlarged lateral ventricle give the tumour a scalloped look defined as the “scalloping sign”, a characteristic radiological feature of this tumour entity [19]. Punctate calcifications are quite common and best seen on computed tomography (CT) scans. Hemorrhage is typically found in larger tumourous lesions and sometimes presents as the fluid-fluid levels in the intratumoural cysts [18] Monomorphous neurocytic tumour cells have round nuclei with salt-and-pepper chromatin and their neurites intermingle into a fibrillary background. Immunohistochemistry shows positivity for NeuN, MAP2 and synaptophysin, while GFAP, OLIG2, and IDH1 R132H staining are negative [20]. Histology shows no or few mitoses. The MIB-1 index is usually low. Higher indexes (>3%) could indicate a tumour with more aggressive behaviour.

Surgery is the first-line treatment with an attempt of complete resection. Radiotherapy is often proposed in case of incomplete resection or recurrence, either stereotactic radiosurgery (SRS) or standard radiotherapy, providing good local tumour control and improved survival [21], even discussing craniospinal irradiation associated with adjuvant chemotherapy in case of dissemination [22]. Chemotherapy, although its place is less well defined, appears as a salvage option in relapsing cases: (i) Temozolomide alone or combined with radiotherapy [23], (ii) Lomustine [24] and (iii) Etoposide/cisplatin/cyclophosphamide [25]. Overall, relapses after surgery and/or radiotherapy are quite rare. The overall survival rate at 10 years is 80% [26].

### 2.3. Extraventricular Neurocytoma

Extraventricular neurocytoma is a WHO grade-2 neoplasm found anywhere in the CNS and outside the ventricular system, affecting young adults with a median age of 30 years. This entity is rarer than central neurocytoma. Seizures and headaches are the most common symptoms revealing the disease [27].

Radiologically extraventricular neurocytomas present as polymorphous large intraaxial lesions with, typically, a mixture of solid and cystic tumour parts, a heterogeneous contrast enhancement and usually an absence of peritumoural oedema [28] as well as calcifications. Half of these lesions are either in the temporal or frontal lobes [29]. Extraventricular neurocytomas have a wide range of histological aspects with glioneuronal phenotype that may resemble ganglioglioma or even oligodendroglioma in case of marked calcifications. Rare cases of histological signs of aggressiveness have been documented. *FGFR1*:*TACC1* fusion is present in two-thirds of cases [30].

Generally, complete surgical resection is the first-line treatment with good overall survival. However, high recurrence rates have been reported in the case of atypical neurocytomas (histology showing increased cellularity, neovascularization and/or necrosis) with partial resection. Adjuvant radiotherapy can be used in case of subtotal resection [28]. Chemotherapy use is anecdotic.

## 3. Glioneuronal Tumours

A brief list of some of the rarer glioneuronal tumours as well as their characteristics is represented in Table 1.

**Table 1 cancers-15-01120-t001:** Main characteristics of some rare predominantly glioneuronal and neuronal tumours (WHO grade 1 and 2).

Tumour	Median Age (Years)/ Sex	Localisation	Immunohisto Chemistry	Specific Mutations	Histology	Symptoms	Treatment	MRI Particularities
Multinodular and vacuolating neuronal tumours	40 y, M > F [31]	Temporal/frontal lobes,	OLIG2+, synaptophysin-, CD34-, GFAP-. NeuN- [32]	MAPK pathway mutations of MAP2K1 and of BRAF (excluding BRAF^V600E^) [33]. FGFR2 fusion	Purely neuronal (non-neurocytic and no neoplastic glial cells). Absence of mitoses.	Mainly seizures [34] Sometimes incidental finding.	Observation preferred. Surgery only if refractory epilepsy [35]. Molecular targeted therapy *.	Small superficial cortical cystic lesions, sometimes in clusters [31].
Rosette-forming glioneuronal tumour (Figure 2)	20 y	Midline structures in proximity of the 4th ventricle and the aqueduct of Sylvius	Neurocytes: Olig2+, rosettes: synaptophysin+, Glial cells: GFAP+, S100+	*FGFR1* mutations are very common, associated with *PIK3CA, PIK3R1* or *NF1* mutations [36]	Biphasic tumour with a component of neurocytes forming rosettes and/or pseudorosettes, and a glial component (often pilocytic astrocytes).	Progressive brainstem/cerebellar signs and visual disturbance.	Surgical resection is preferred [37]. If aggressive features and/or leptomeningeal infiltration, spinal metastasis: RT and chemotherapy [38] can be discussed in an adjuvant manner after surgery. Molecular targeted therapy*.	Mix of cystic and solid lesions, strong gadolinium enhancement. “Green bell pepper sign” [39].
Papillary glioneuronal tumour	25 y	Supratentorial: mainly temporal and frontal lobes	Neurocytes: Olig2+, synaptophysin+, Astrocytic cells of papilla: GFAP+	*PRKCA* gene fusions, mostly *SLC44A1*:*PRKCA* fusion [40,41]	biphasic organisation with astrocytic papillas around hyalinized vessels and a neuronal component (most often neurocytic).	Headaches [42] and seizures. Incidental finding if small enough. Often characterized by an indolent course [43].	Surgical resection alone is preferred: with very rare recurrences [44]. RT and/or chemotherapy if high Ki-67 in recurrence or other features of aggressiveness [45].	Solid and a cystic component [46]. Septations can be quite specific. Calcifications are frequent.
Myxoid glioneuronal tumour (previously DNT of the septum pellucidum)	20–25 years	Septum pellucidum, periventricular locations, corpus callosum [47].	OLIG2+, SOX10+, GFAP+	dinucleotide mutation at codon p.K385 in the *PDGFRA* gene [47]	Histologically similar to DNT or RGNT.	Hydrocephalus the most frequent initial clinical presentation, incidental findings not rare.	Gross surgical resection alone is preferred. In case of relapse and/or dissemination, RT [48] and/or chemotherapy (TMZ, CCNU) [47] can be considered.	No contrast enhancement, nor diffusion restriction. Partially suppressed Flair in centre, no oedema [48,49].
Gangliocytoma	Children, young adults	Mainly temporal lobe [50]. Sellar locations also seen [51].	chromogranin A+, synaptophysin+, neurofilament+, GFAP-	BRAF^V600E^ mutation of alternative MAPK pathway alterations [52].	multinucleated ganglionic neuronal tumour cells.	Seizures due to temporal/cortical locations [50,53]. Sometimes headaches, brainstem signs.	Surgical resection [53]. Relapse after resection remains very rare [54]. Chemotherapy has no place, neither radiotherapy. Molecular targeted therapy*.	Strong Gd enhancement, cystic images, perilesional oedema, calcifications [54].
Diffuse glioneuronal tumour with oligodendroglioma-like features and nuclear clusters	Young adults, children	Supratentorial locations.	Synaptophysin+, NeuN+, MAP2+, Olig2+, GFAP-	Monosomy 14. Distinct methylation profile [55].	Pseudo-oligodendroglial cells infiltrating cerebral cortex and forming nuclear clusters. Low number of mitosis.	Unspecific.	No standard treatment. Gross total surgery followed by radiotherapy may be a good option [56].	None or only discrete Gd enhancement, no oedema.

Molecular targeted therapy *: Molecular targeted therapy is an option if druggable molecular alteration is detected (e.g., BRAF or FGFR inhibitor).

### 3.1. Ganglioglioma

Ganglioglioma (Figure 3) is the most frequent glioneuronal tumour. It represents 0.5–1% of all primary CNS tumours and affects mainly young patients. Gangliogliomas are mostly located in the temporal lobe. These lesions are very epileptogenic and frequently associated with drug-resistant epilepsy. Comorbid chronic psychosis and epilepsy can be associated in the case of amygdala ganglioglioma [57]. Associated seizures can also present as “panic attacks” in young patients with ganglioglioma located in the temporal or cingulate gyrus areas [58,59]. These cases, as well as their locations, are quite atypical.

Some initial presentations are also atypical. Indeed, in a cohort of 14 gangliogliomas, 3 had hemorrhagic presentations [60]. Other atypical and rare locations, including the optic pathways [61] and the spinal cord (higher risk of recurrence) [62], have been reported. MRI shows a solid mass with a cystic and/or calcified component in up to 50% of cases. The lesion is hypo- or isointense on T1-weighted images and hyperintense on T2-weighted images with variable contrast enhancement [63]. Histology shows ganglionic neuronal tumour cells, which express synaptophysin and/or Chromogranin A and which can be binucleated by contrast with residual normal neurons. It also contains glial tumour cells expressing OLIG2 and/or GFAP and presenting various aspects: most often piloid astrocytic but also oligodendroglial differentiation. The Ki67 proliferation index is usually < 3%. The tumours frequently contain eosinophilic granular bodies, a significant lymphocytic infiltrate, and CD34 immunopositive stellar cells.

Complete surgical resection is the first-line treatment with an overall survival rate at 10 years > 80%. GTR (gross total resection), when achievable, was shown to increase OS, the temporal location of the tumour has a better prognostic factor than infratentorial locations [64], GTR allows a better seizure control compared to STR [65]. Brainstem low-grade gangliogliomas with maximum achievable surgical resection followed by observation without immediate adjuvant therapy could also be a safe strategy [66]. Although the role of radiotherapy is debated, it is commonly used after: (i) partial resection of aggressive high-grade tumours, (ii) partial removal of low-grade tumours located in eloquent areas such as the brainstem and, (iii) tumour relapse, especially as a salvage treatment [67,68,69]. Chemotherapy is less studied for treatment of ganglioglioma patients. It can be proposed to patients with anaplastic features [70] or relapsing tumour not eligible for radiotherapy or surgery. Various regimens have been suggested such as Temozolomide in adults [71,72] and Carboplatin in pediatric cases [73]. There is little evidence that chemotherapy significantly increases OS.

BRAF^V600E^ mutations, encountered in ~30% of gangliogliomas, are correlated with a worse prognosis [74]. Nonetheless, BRAF^V600E^ mutations are actionable using MAPK signaling pathway inhibitors with clinical benefit [75]. Tumours without BRAF^V600E^ mutation can be characterized further to identify alternative druggable genetic alterations of MAPK signaling pathway (e.g., *FGFR1* missense mutation, *BRAF* fusion).

“Anaplastic ganglioglioma” accounts for less than 5% of all gangliogliomas and was previously considered WHO grade 3 and associated with increased seizure frequency, although the new WHO 2021 classification no longer recognizes it as a distinct entity. A methylome profiling of the tumour can identify an alternative diagnosis to ganglioglioma in such a context. Only about a quarter of cases are of temporal location. Histology shows usually more than 10 mitoses per mm^3^, necrosis and microvascular proliferation. The median OS is about 2 years [76].

The optimal treatment would be complete surgical resection followed by radiotherapy [77]. A Stupp protocol can be proposed as a first-line of treatment after surgery [78] adjuvant radiotherapy with chemotherapy in case of STR [77]. In case of relapse, targeted molecular therapies including BRAF inhibitor alone or combined with MEK inhibitor, can be proposed [78,79,80] (combination of both can overcome BRAFi resistance) [81].

### 3.2. Dysembryoplastic Neuroepithelial Tumour (DNET)

DNET is a rare primary WHO grade-1 CNS tumour most often revealed by severe focal epilepsy in young patients aged between 10 and 25 years. It is located in the temporal lobes in two third of cases.

Young patients with DNET may also present psychiatric problems in addition to temporal epilepsy, which can be improved by surgery [82].

MRI shows well-demarcated hypointense T1-weighted and hyperintense Flair/T2-weighted lesions with multicystic components and high ADC value as an expression of a low cellular density. Perilesional edema and mass effect are usually lacking, and contrast enhancement is rare (20–30% of cases). Perfusion shows a lower relative cerebral blood volume (rCBV) in comparison to the surrounding cerebral parenchyma. Focal cortical dysplasia is frequently associated [83,84]. Three histologic subtypes depending on the size of the glial and neuronal compartments have been described: (i) simple, (ii) complex and, (iii) diffuse. DNET are characterized by the glioneuronal specific element, an architecture made of parallel axes separated by a myxoid extracellular matrix containing “floating neurons”. The parallel axes are composed of monomorphous pseudo-oligodendroglial tumour cells associated with a fibrillary background and grouped around a capillary or a bundle of axons. Immunohistochemistry shows positivity for NeuN and synaptophysin of the floating neurons and Olig2 of the oligodendroglial cells. Biology shows *FGFR1* alterations in more than 75% of all DNT and BRAFV600E mutations in less than one-third of cases [85].

Although these tumours are very rarely aggressive, treatment is sometimes necessary in symptomatic cases (e.g., severe epilepsy). Surgery is the cornerstone of treatment. Complete surgical resection allows a better clinical outcome [86]; more than 80% patients are seizure free at 1 year, especially in case of GTR and shorter epilepsy duration [87]. Radiotherapy and chemotherapy have no place in DNET. Malignant transformation is also very rare, only seen in complex type DNET and extratemporal locations [88].

### 3.3. Diffuse Leptomeningeal Glioneuronal Tumour (DLGNT)

DLGNT is a glioneuronal neoplasm that commonly involves diffusely the leptomeninges but circumscribed intra-axial unifocal presentations also exist in adults [89]. Less than 100 cases have been reported [90]. DLGNT mainly affects children and males. Clinical signs are unspecific, depending on tumour location, including increased intracranial pressure, focal neurological deficit, cerebellar syndrome, and hydrocephalus.

MRI reveals most often thickened leptomeninges with a corresponding nodular leptomeningeal contrast enhancement and small subpial cysts. Discrete parenchymal lesions and ventricular nodules may be present. Differential diagnoses, including infectious (e.g. tuberculosis), inflammatory and carcinomatous meningitis, should be ruled out first. CSF analysis does not show each time tumour cells, however a high protein count is often seen [91]. Communicating hydrocephalus may be observed on an MRI [92]. Histomolecular examination shows oligodendrocyte-like tumour cells positive for OLIG2, Synaptophysin, MAP2 and S100 but negative for IDH1 R132H. The proliferation index is most often low. Chromosome arm 1p deletion (or 1p/19q codeletion) and MAPK signaling pathway activation through various genetic alterations, including BRAF fusions (e.g., *KIAA1549-BRAF*), are the most common molecular events. Two subtypes are defined based on methylation classes DLGNT-1 and DLGNT2. DLGNT2 and/or gain of 1q are associated with a worse prognosis [93]. Ki-67 > 7% was associated with poorer OS [90].

Due to its scarcity, no standard of care is established. Since DLGNT are most often slow-growing tumours with a limited number of anaplastic and aggressive cases, a wait-and-monitor strategy is often performed [94]. When treatment is needed, multiple options have been suggested: (i) radiation therapy with various schemes [95] or (ii) single or multiple-agents chemotherapy, Temozolomide is usually preferred [95,96] as well as Bevacizumab [97], and was shown to increase OS in pediatric cases [98]. Surgery has little place for this extensive disease and is limited to biopsy sampling, removal of a symptomatic node, and management of hydrocephalus.

Recurrent 1p deletion and MAPK/ERK pathway activation could represent, in theory, a potential therapeutic target, e.g., for MEK inhibitors [93].

### 3.4. Cerebellar Liponeurocytoma

Cerebellar liponeurocytoma is a very rare (less than 100 reported cases) WHO grade-2 primary CNS tumour located in the posterior fossa (more often in the cerebellar hemispheres than in the vermis), affecting adults aged ~50 years. It has the same clinical presentation as other low-grade cerebellar tumours [99].

This type of tumour is easily identified on an MRI if macroscopic fat is detectable. MRI usually shows a hypointense T1-weighted lesion that may present with patchy hyperintense areas corresponding to regions of fat, a hyperintense signal on Flair/T2-weighted sequences and a heterogeneous contrast enhancement. Little or no perilesional oedema can be found [100]. Lipidization is not pathognomonic of liponeurocytoma. Rare cases of other primary CNS tumours with small lipid content have been reported (e.g., medulloblastoma, cerebellar astrocytoma, ependymoma) [101]. Histology shows a well-delimitated tumour with a strong neurocytic component arranged in lobules, variable glial differentiation, and areas of lipidization within neuroepithelial tumour cells resembling adipocytes. Immunohistochemistry is usually positive for NeuN, synaptophysin and MAP2. Olig2 is usually absent. Ki67 index is usually low [102].

The first line of treatment is surgical resection. No standard treatment exists. One-third of operated patients relapse. Radiotherapy is an option, especially in case of relapse and for incomplete resection: recurrence rates is less frequent after adjuvant radiotherapy than surgery alone [100], although other authors did not find a significant increase of PFS after post-operative radiotherapy, with GTR being the only significant factor allowing longer PFS; chemotherapy is anecdotic (1 case described in the literature) [103]. The overall survival rate is about 70% at 10 years.

### 3.5. Subependymal Giant Cell Astrocytoma (SEGA)

SEGA (Figure 4) is a glioneuronal tumour, WHO grade-1, occurring almost exclusively in young patients with tuberous sclerosis (10–15% of all tuberous sclerosis patients) [104], but sporadic SEGA in the absence of tuberous sclerosis also exist [105]. TSC occurs in approximately 1 in 6000 births worldwide. TSC are characterized by *TSC1* or *TSC2* mutations, which are responsible for the overactivation of the mTOR signaling pathway, leading to aberrant cell development [106] TSC may manifest with 3 principal intracranial pathological entities: cortical tubers, subependymal nodules (SENs), and SEGAs.

Seizures are quite often seen in this population of patients.

SEGAs are diagnosed either due to obstructive hydrocephalus with a blockade of the foramen of Monro or during systematic MRI monitoring in patients harbouring a tuberous sclerosis. There are several factors that may accelerate the growth of SEGAs: the size of tumour > 2 cm, younger age of patient, and TSC2 genotype [107].

These lesions at the caudothalamic groove with either a size of more than 1cm in any direction or a subependymal lesion at any location that has shown serial growth on consecutive imaging regardless of size [107]. MRI shows typically a hyperintense lesion on FLAIR and T2-weighted images with a strong contrast enhancement (most of the time, but not always) after intravenous gadolinium-based contrast media application [108]. Calcifications may be present.

Histology shows astrocytic cells with large cytoplasm. Immunostaining for GFAP and TTF1 are positive. Neuronal markers such as synaptophysin and neurofilament may also be expressed. The presence of mitoses, necrosis and microvascular proliferation has no adverse prognostic value.

Surgery is preferred in case of hydrocephalus or signs of elevated intracranial pressure. mTOR inhibitors (mTORi) (e.g., everolimus) have become the first-line treatment for the management of SEGAs not requiring immediate surgical treatment, with studies showing a strong volume contraction as a result of the treatment and subsequent hydrocephalus prevention, as well as seizure reduction and even improvement of other manifestations of TSC [109,110]. Due to the risk of tumour regrowth in case of treatment discontinuation [111], a maintenance therapy mTORi may be needed. The overall incidence of AEs with mTORi is 30–74% (bronchitis, stomatitis, pyrexia), and 7% of patients discontinued the treatment [111,112]. The EMINENTS study showed that a low-dose Everolimus maintenance therapy is as effective with fewer AEs than the standard dosage [113]. mTORi discontinuation has been associated with seizure relapse [114].

Yearly MRI surveillance is recommended in young tuberous sclerosis patients to begin treatment at an early stage in case of SEGA occurrence. Radiotherapy is used much less frequently than before the introduction of mTORi and classical chemotherapy has no place in the therapeutic strategy. OS at 5 years is excellent in most cases [110].

## 4. Glial Tumours

### 4.1. Pilocytic Astrocytomas

Pilocytic astrocytoma (PA) (Figure 5) is a slow-growing, well-circumscribed, WHO grade-1, glial tumour. It is mostly diagnosed in children (the most frequent primary CNS tumour in children and adolescents, especially in children with NF-1, Noonan syndrome and tuberous sclerosis), with adult cases being 10 times less common. In adult patients, they can be found both in supratentorial and infratentorial locations and have a worse prognosis than in pediatric patients [115]. A total of 27% of adult PAs occur in the cerebellum and 30% in the cerebrum/lobar localization; 90% of all cases harbour MAPK pathway abnormalities [115,116].

Due to slow growth, the symptoms usually evolve in a very progressive manner. Sudden and hemorrhagic presentations are rare: they are seen in adult patients (median age 37y) with supratentorial and hypothalamic/suprasellar tumours [117]. In younger patients, psychiatric manifestations, such as eating disorders, behaviour changes and psychosis, may be encountered [3].

On an MRI, the typical appearance involves a large cystic component with an enhancing mural nodule in 2/3 of cases. The solid component has usually a hypointense signal on T1-weighted images and a hyperintense signal on T2-weighted images with a homogenous contrast enhancement. When occurring in the optic-diencephalic/brainstem location, PAs present as infiltrative solid masses with a fusiform enlargement of the optic nerves and variable contrast enhancement. A tectal location may result in hydrocephalus due to aqueductal obstruction. Spinal PAs are usually well-defined and often associated with syringomyelia and large cystic components [118]. NF1 is associated with an increased risk of glioma, and PAs represents about half of all NF1-associated gliomas. Roughly 15% of NF1 patients have pilocytic astrocytoma, particularly in the optic pathway. PAs are associated with abnormalities in gene-encoding members of the MAPK signaling pathway, one of the most frequent being the *KIAA1549*-*BRAF* fusion, resulting in constitutive activation of BRAF kinase activity. Histology shows low-to-moderate cellularity with compact, densely fibrillated areas consisting of cells with long bipolar hair-like (pilocytic) processes, as well as loosely textured areas, composed of multipolar cells. An additional oligodendroglial tumour component may also exist, resulting in the classical biphasic aspect. Immunohistochemistry is positive for GFAP, OLIG2 and S-100 [115]. Ki67 is usually 1–5%. Cases of histological signs of aggressiveness are very rare and incite to rule out a HGAP, especially in adults and in NF1 patients.

The first-line treatment is surgery, preferably complete surgical resection (including cerebellar [119] and spinal forms [120]), followed by observation [115,121]. GTR reduces the risk of recurrence compared to STR (27% vs. 73%) [121]. No study has confirmed a clear benefit of adjuvant chemotherapy/radiotherapy after a surgical resection, with two major studies finding a negative role of RT in the management of PAs concerning survival, but all RT regimens were considered altogether [122,123]. Radio-induced tumours have been described several years after pediatric PAs irradiation [124].

Stereotactic radiotherapy/radiosurgery (SRT/SRS) could be proposed in adult patients and in the case of midline/brainstem lesions: SRT was shown to be effective in controlling residual PAs without serious side effects [125], effectively improving PFS but not OS in adult patients [126]. A multicentric retrospective study showed favourable long-term PFS and OS in patients with PAs after treatment with Gamma Knife SRS, either in a first-line setting or as a salvage treatment [127], and another study showed high rate of pseudoprogression cases within 12 months after SRT [128]. A review concluded SRS is a safe and promising therapeutic in PA management [129] and should be discussed case by case.

Chemotherapy (CT) is a valid choice in optic gliomas or in pediatric populations with local recurrence [116]. Leptomeningeal dissemination is very rare [130] but exists, and can mimic a DLGNT at initial diagnosis. Bevacizumab was shown to induce a durable response in recurrent PA [131,132], Temozolomide can also be discussed [133].

MEK1/2 inhibitors target aberrant over-activation of the MAPK pathway: selumetinib treatment showed responses and prolonged disease stability in patients with WHO grade-1 PA with either a KIAA1549-BRAF fusion or the BRAF^V600E^ mutation [134,135], as well as Trametinib in infants in case of CT failure [136,137]. Dabrafenib is also efficient in controlling BRAF^V600E^ PAs [138]. A case report showed a 19-month response in an adult patient with recurrent PA after Pemigatinib (pan-FGFR1) treatment [139].

### 4.2. High-Grade Astrocytoma with Piloid Features (HGAP)

HGAP is a very rare high-grade (grade 3–4) primary CNS tumour, recently highlighted due to its DNA methylation profile. It is located mainly in infratentorial and spinal locations and primarily affects adult patients with a median age of 41 years.

Radiologically, they appear hyperintense on T2-weighted and hypointense on T1-weighted images, show a tendency to rim enhancement after contrast media application and are often surrounded by peritumoural oedema. Histology shows an astrocytic tumour of various aspects, from circumscribed to diffuse neoplasms and from pilocytic astrocytoma with increased mitotic activity to glioblastoma-like aspects. Molecular biology shows the absence of *IDH* mutations, the frequent presence of *CDKN2A/B* homozygous deletion and MAPK pathway mutations (KIAA1549-BRAF fusion, *NF1* inactivating mutations, *FGFR1* activating mutations), and in half of cases, loss of ATRX expression [140].

Due to its scarcity, there is no standard treatment. Maximal resection is usually performed, followed by radiotherapy with adjuvant chemotherapy (Temozolomide). The Charité series reported a patient with a stable disease for a few months under anti-MEK treatment binimetinib [140,141].

### 4.3. Astroblastoma MN1 Altered

Astroblastoma MN1-altered (Figure 6) is a circumscribed glial supratentorial neoplasm affecting primarily young women between 10 and 30 years. Clinically, astroblastomas have similar symptoms as other slow-growing brain tumours. This lesion is predominantly found in the frontal and parietal regions.

On an MRI, astroblastomas appear as well-delimitated nodular tumours with heterogeneous contrast enhancement and cystic components resulting in the characteristic bubbly appearance. Commonly calcifications and hemorrhage are present and perilesional edema can be found [142]. The main histological feature of astroblastoma is astroblastic pseudorosettes with perivascular hyalinization and fibrosis. Immunohistochemistry shows positivity for GFAP, Olig2, S100, EMA, and podoplanin.

Astroblastoma are molecularly characterized by the rearrangements of the *MN1* gene at chromosome band 22q12.1 and gene fusion involving *MN1* [143,144]. Prognostic factors for astroblastoma MN1-altered are poorly described and this tumour type has no CNS WHO grade according to the latest classification.

Surgical resection is the first-line treatment both in low-grade and high-grade astroblastomas, followed by radiotherapy in case of aggressiveness or early recurrence. Due to its rare nature, no standardized treatment exists. The place of chemotherapy is debated. Overall, the 10-year survival rates are more than 50% [145] (p. 1).

### 4.4. Chordoid Gliomas

Chordoid glioma is a WHO grade-2 tumour located in the anterior part of the third ventricle, mostly seen in middle-aged adult women (ratio F/M 2/1). Due to its location in the third ventricle, symptoms include headache, visual disturbances due to proximity to optic structures [146], cognitive alterations, and gait difficulties. Other signs may be related to hypothalamic/pituitary dysfunction due to tumour extension and SIADH [147,148].

On an MRI, this tumour forms an isointense round mass on T1-weighted images and slightly hyperintense mass on Flair and T2-weighted images, with homogeneous and intense contrast enhancement; MRI spectrum showed an elevated choline value and reduced N-acetylaspartate value [149]. Chordoid gliomas are usually well-demarcated on an MRI with cystic components in about a quarter of cases [150]. Histology shows a chordoid architecture made of rows of glial cells separated by a myxoid extracellular matrix. Immunohistochemically revealed GFAP, vimentin, TTF1, CD34 and EMA-positive tumour cells. There is a low Ki67 index. The p.D463H missense mutation in the *PRKCA* gene is typical [151].

Surgical resection is the first-line treatment [149]; GTR is associated with excellent tumour control [152]. However, due to its proximity and extension towards the hypothalamus and optic chiasma, there is a risk of sequelae (e.g., hypopituitarism), thus surgery is often subtotal or partial. In the case of complete resection, the overall survival is excellent. In the case of partial resection, some centres use complementary radiotherapy and/or radiosurgery with good results [153,154,155]. Chemotherapy is not used [156].

### 4.5. Pleomorphic Xanthoastrocytoma (PXA)

PXA (Figure 7) is a rare astrocytic tumour (grade 2, sometimes 3) mostly found in young adults (median age: 29 years). They are almost always found in supratentorial locations (mainly temporal lobe), although very rare infratentorial/spinal and even retinal locations have been described [157]. Most patients suffer from drug-resistant seizures. [158].

MRI shows a well-delineated superficial mass, most often comprising solid and cystic components with vivid contrast enhancement. Adjacent leptomeningeal contrast enhancement may be present. Histology shows large pleomorphic cells, spindle cells, and lipidized cells surrounded by a reticulin network of extracellular matrix; there is no necrosis as well as few mitoses in grade 2. Immunohistochemistry shows positivity for GFAP, S100, vimentin, as well as scarce and inconstant positivity for neuronal markers such as synaptophysin or neurofilament. In the case of grade 3, there are > 5 mitoses and possibly microvascular proliferation and necrosis. Almost 70% of all PXAs have BRAF mutations. Targetable mutations, such as *BRAF* p.V600E, are the most frequent. In tumour without BRAF^V600E^ mutation, extensive screening is valuable to identify an alternative druggable genetic alteration of the MAPK pathway (e.g., gene fusion involving *BRAF, RAF1, ALK, ROS1*). *CDKN2A/B* loss is also very frequent, up to 90% [158].

Surgery is the cornerstone of treatment, especially complete surgical resection if possible, associated with radiotherapy with doses up to 54 Gy in case of incomplete resection or signs of aggressiveness.

A meta-analysis with patients having low-grade gliomas, including gangliogliomas, astroblastomas and xanthoastrocytomas, showed that early radiotherapy increased time to progression and allowed better seizure control, although there was no difference in overall survival when compared to delayed radiotherapy [159].

Stereotactic radiotherapy can also be used effectively in case of residue [160]. Malignant progression is more often seen than in other grade-2 tumours. In case of unresectable tumour progression, chemotherapy (especially Temozolomide [133]) can be used, especially if new surgery is not possible with variable effectiveness. In the case of leptomeningeal dissemination, craniospinal irradiation associated with chemotherapy can be used. The VE-Basket study has shown that BRAF-mutated PXA can respond quite well to molecular-targeted therapy (BRAF inhibitors) [6]. One review found an average OS of 193 months, with older age, post-operative RT and a larger tumour size being associated to a worse OS [161], while another found a median OS of about 34,9 months, with an OS of 50% after 5 years and recurrence being a risk factor for decreased OS [162]. Bevacizumab treatment was reported only in one case-report of anaplastic PXA, with a short-term response [163]. Older age and a larger > 3 cm tumour size at diagnosis are risk factors for poor OS. Grade-3 WHO have a shorter OS than grade 2.

## 5. Conclusions

Rare glial and/or neuronal tumours are a heterogeneous group of mainly slow-growing primary CNS tumours. Thanks to the new WHO 2021 classification, these tumours have been better characterized on a histological and molecular biology level, with fewer cases of undetermined histology. Methylome technics may allow in the future to further improve classification and even determine the prognosis of subtypes of glial and/or neuronal tumours according to the methylation class [164].

MRI is the radiological gold standard for diagnosing and disease surveillance of glial and/or neuronal tumours.

The cornerstone of the treatment of symptomatic rare glial and/or neuronal tumours is maximum safe surgical resection: gross total resection is recommended if feasible with excellent OS, even in the case of grade-2 WHO tumours: surgery allows better symptom control, especially in the case of associated epilepsy [165]. In most grade-1 WHO tumours, overall survival rates are excellent, more than 70–80% at 5 years. Radiation therapy can be proposed in grade-2 tumours with a subtotal resection and early relapse.

Temozolomide regimen can be proposed in case of DLNGT or non-operable grade-2 WHO tumours.

*BRAF* alterations, missense mutations or aberrant fusions, as well as other MAPK alterations, are seen in numerous glioneuronal tumours. Targeting the MAPK pathway is difficult in most solid cancers, including CNS tumours, even if mutations are present, because of the occurrence of drug resistance over time [33]. Another gene family alterations found in many CNS neoplasms are FGFR alterations, especially fusions, which could constitute an interesting target for FGFR inhibitor drugs in future trials [166].

New research will focus on improving target therapies, such as new BRAF/MEK inhibitors, using FGFR inhibitors, Raf inhibitors, or a combination of multiple therapies, in relapsing rare glial and/or neuronal tumours after surgery or surgically inaccessible tumours.

## Figures and Tables

**Figure 1 cancers-15-01120-f001:**
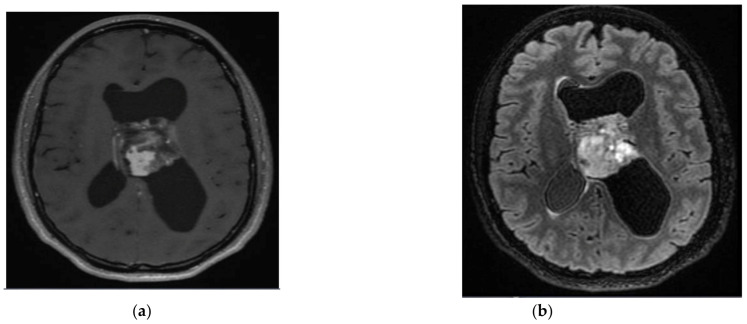
A 29-year-old female’s central neurocytoma. After surgical resection, the patient was monitored without recurrence. (**a**) T1-weighted image after gadolinium infusion; (**b**) Flair-weighted sequence.

**Figure 2 cancers-15-01120-f002:**
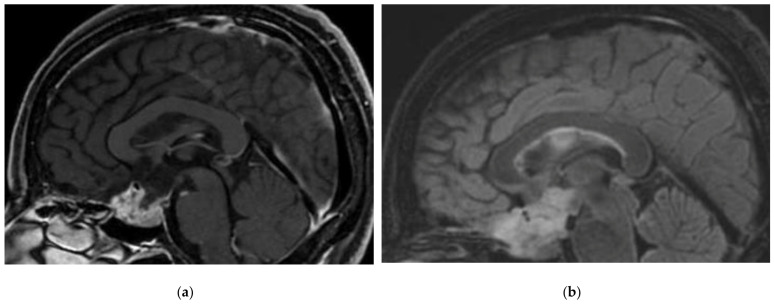
A 40-year-old female with Rosette-forming glioneuronal tumour (RGNT) of hypothalamic region, an incidental finding on an MRI performed for vertigo investigations. No disease activity after subtotal resection. Sagittal views: (**a**) T1-weighted images after gadolinium infusion; (**b**) Flair-weighted sequence.

**Figure 3 cancers-15-01120-f003:**
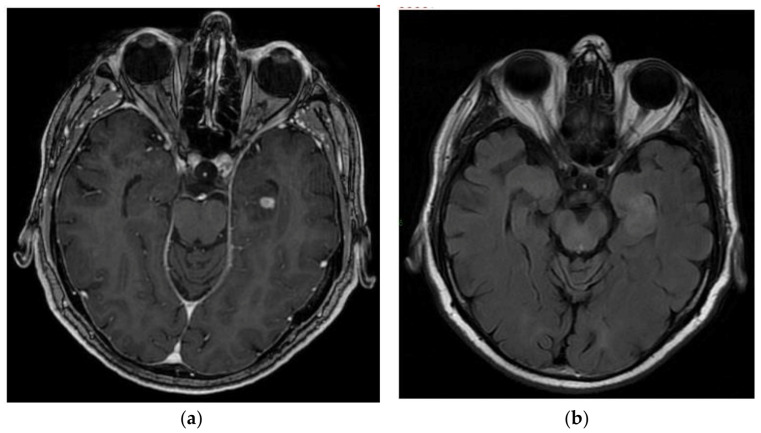
A 72-year-old male with left temporal lobe ganglioglioma. After the diagnostic biopsy, the patient was followed and treated with chemotherapy and radiotherapy at recurrence. (**a**) T1-weighted image after gadolinium infusion; (**b**) Flair-weighted sequence.

**Figure 4 cancers-15-01120-f004:**
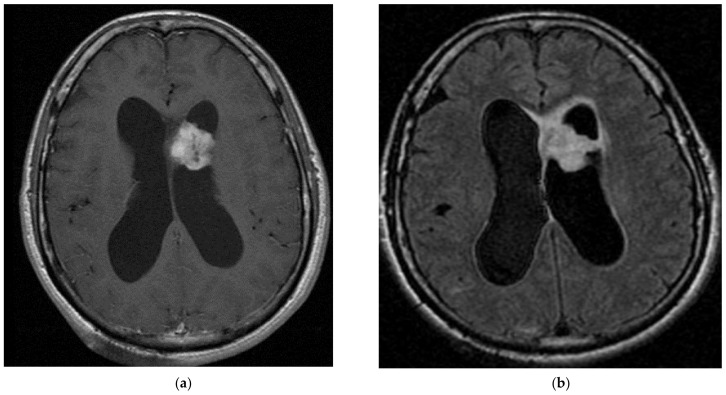
Young male aged 38 years old with subependymal giant cell astrocytoma (SEGA) discovered on a systematic MRI. Due to mTOR inhibitor discontinuation, a slowly progressive disease requiring surgical resection. No disease activity after surgery. Axial view, (**a**) T1-weighted image after gadolinium infusion; (**b**) Flair-weighted sequence.

**Figure 5 cancers-15-01120-f005:**
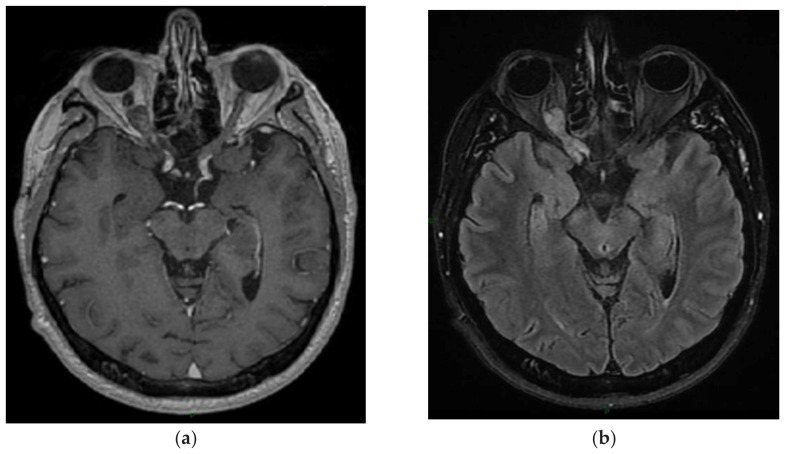
A 59-year-old male with pilocytic astrocytoma of the right optic nerve diagnosed at 49 years. After the diagnostic biopsy, the patient was treated with chemotherapy. (**a**) T1-weighted image after gadolinium infusion; (**b**) Flair-weighted sequence.

**Figure 6 cancers-15-01120-f006:**
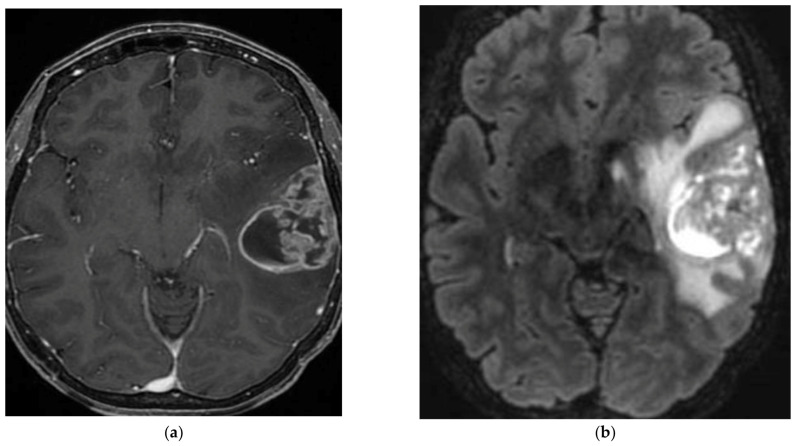
A 44-year-old female with left temporal astroblastoma diagnosed on an MRI after multiple seizures. After total resection, the patient received radiotherapy. (**a**) T1-weighted image after gadolinium infusion; (**b**) Flair-weighted sequence.

**Figure 7 cancers-15-01120-f007:**
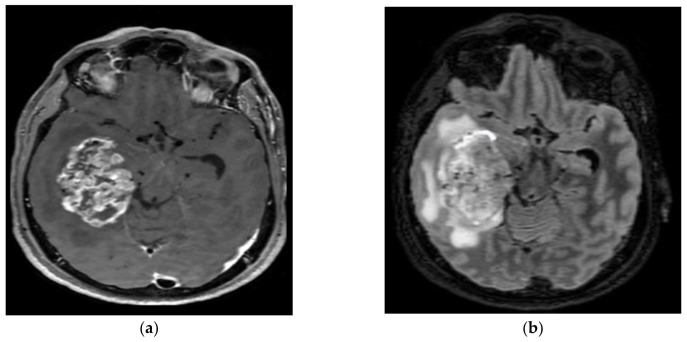
A 60-year-old male with right temporal lobe anaplastic pleomorphic xanthoastrocytoma. After surgical resection, the patient was treated with chemotherapy and radiotherapy at recurrence. (**a**) T1-weighted image after gadolinium infusion; (**b**) Flair-weighted sequence.

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
