# Peer review of "Rare Neuronal, Glial and Glioneuronal Tumours in Adults"

_cancers, 2023, doi:10.3390/cancers15041120_

Round 1

Reviewer 1 Report

The manuscript “Rare neuronal, glial and glioneuronal tumors in adults” by Crainic et al is informative and has certain merits, however, considerable improvement can be made by addressing the following issues:

Major issues:

1.       Images are generally scare or lacking. Addition of images of all, or at least most, tumor types, would add considerable value to this review. Now there is only a few examples of radiological images, MRIs, of selected tumors and the reason why these types of tumors specifically have been selected for illustration is unclear.

2.       References are lacking at numerous places, in particular, and most importantly, references are needed in texts where treatments are described. As an example, just to mention one, in the section on Ganglioglioma, the authors write: A Stupp protocol can be proposed (line 187). The authors need to add the reference for this claim and clarify at what stage of the disease the Stupp protocol can be recommended etc. This is applicable at many places throughout the manuscript including Table 1.

3.       In the 2021 WHO classification of tumors of the CNS, no clear distinction is made between glioneuronal and neuronal tumors. In contrast, the authors try to make this distinction which have resulted in some confusion:  In the Introduction, MVNT is listed as “neuronal”, but MVNT is also included under “glioneuronal tumors” in Table 1. Cerebellar liponeurocytoma is listed under “neuronal tumors in the Introduction but later (line 239) it is listed under “glioneuronal tumors”. Furthermore, some tumor types are described in some detail in the main text and some other tumor types are only briefly overviewed in a table. One might suggest the addition of a new table showing an overview of all tumors and their categorization into predominantly neuronal, glioneuronal or glial lineages. All types of tumors might then be described in the text along with relevant MRIs.

4.       The treatment discussed for ganglioglioma differs significantly between table 1 and text (lines 166-179). In general, for many of the tumor types discussed, treatment options are better discussed in the text (with references!) than very briefly and imperfectly in a table.

5.       Although gangliocytoma occur predominantly in the pediatric population it is also diagnosed in young adult patients, and it would be appropriate to include also this type of tumor in this review of rare tumors in adults.

Further general comments:

1.       A summary of how the 2021 classification rare neuronal, glial and glioneuronal tumors in adults differ from previous WHO classification would be informative

2.       A language revision by a native English speaker is recommended.

Author Response

Paris, December 29th, 2022

Ref.: Ms. No. cancers-2104777

Dear Editor in Chief,

            You will find enclosed a revised version of our manuscript (Ms. No. cancers-2104777) entitled “Rare neuronal, glial and glioneuronal tumors in adults”.

            In this revised version, we have taken into consideration all the comments made by the referees. The responses to each comment are marked in bold in the present letter and changes have been highlighted using the tracking system of word software in the revised manuscript.

            We thank the reviewer for his helpful comments. These comments have permitted a significant improvement of our manuscript which, we believe, should now be suitable for publication in Cancers.

Sincerely yours,

Ahmed Idbaih, MD, PhD

Reviewer #1

Point #1.1. Images are generally scare or lacking. Addition of images of all, or at least most, tumor types, would add considerable value to this review. Now there is only a few examples of radiological images, MRIs, of selected tumors and the reason why these types of tumors specifically have been selected for illustration is unclear.

Response #1.1: As requested by the reviewer#1, additional images have been added to the manuscript.

Point #1.2. References are lacking at numerous places, in particular, and most importantly, references are needed in texts where treatments are described. As an example, just to mention one, in the section on Ganglioglioma, the authors write: A Stupp protocol can be proposed (line 187). The authors need to add the reference for this claim and clarify at what stage of the disease the Stupp protocol can be recommended etc. This is applicable at many places throughout the manuscript including Table 1.

Response #1.2. We have updated the references as recommended by the reviewer#1. The stage of the disease for Stupp protocol has been specified.

Point #1.3. In the 2021 WHO classification of tumors of the CNS, no clear distinction is made between glioneuronal and neuronal tumors. In contrast, the authors try to make this distinction which have resulted in some confusion: In the Introduction, MVNT is listed as “neuronal”, but MVNT is also included under “glioneuronal tumors” in Table 1. Cerebellar liponeurocytoma is listed under “neuronal tumors in the Introduction but later (line 239) it is listed under “glioneuronal tumors”. Furthermore, some tumor types are described in some detail in the main text and some other tumor types are only briefly overviewed in a table. One might suggest the addition of a new table showing an overview of all tumors and their categorization into predominantly neuronal, glioneuronal or glial lineages. All types of tumors might then be described in the text along with relevant MRIs.

Response #1.3: Indeed, there is no clear distinction in the WHO classification. Therefore, we have renamed entities as predominantly glial, glioneuronal or neuronal tumors across the entire manuscript including the Table 1.

The Table 1 was created to cite the entities not described in the text. Basically to be concise and comprehensive. For more details, references have been added to the Table 1.

Point #1.4. The treatment discussed for ganglioglioma differs significantly between table 1 and text (lines 166-179). In general, for many of the tumor types discussed, treatment options are better discussed in the text (with references!) than very briefly and imperfectly in a table.

Response #1.4: We agree with the reviewer. We have specified the treatments in the Table 1 and we have the references in the Table 1 as well.

Point #1.5: Although gangliocytoma occur predominantly in the pediatric population it is also diagnosed in young adult patients, and it would be appropriate to include also this type of tumor in this review of rare tumors in adults.

Response #1.5. According the recommendations of the reviewer #1, “Gangliocytoma” is now detailed in the text and the Table 1. Actually, in the Table 1, “ganglioglioma” was written instead of “gangliocytoma” by mistake.

Point #1.6. A summary of how the 2021 classification rare neuronal, glial and glioneuronal tumors in adults differ from previous WHO classification would be informative

Response#1.6. We have added a short paragraph at the beginning of the article detailing this classification. “In contrast to previous versions, the new World Health Organization (WHO) classification of primary CNS tumours, pub-lished in 2021, is based on histological features and/or molecular characteristics, such as specific genetic alterations and DNA methylation classes. Indeed, some tumor types need methylation analysis for accurate diagnosis (e.g. HGAP, DGONC)”

Reviewer 2 Report

General comments:

1. The focus on methylation in the Introduction feels misplaced. If the authors wish to begin the review with a comment on methylation profiling of these rare tumours, I would suggest placing this discussion in a free-standing section after the introduction.

2. I would suggest that the figures be revised to include a radiographic example of each tumour reviewed in the text.

3. The manuscript is in need of some editorial attention for occasional lapses in grammar and language usage. For example, in section 2.2, “There have been reported hundred cases in the literature.”

Focused comments”

1. In section 2.1, a brief description of Cowden disease would be valuable.

2. In section 2.2, it is typical to use “hemorrhage” in the singular.

3. In section 3.4, I would ask the authors to speak to duration of mTOR inhibitor therapy and the concern of tumour recurrence with cessation of therapy.

4. The manuscript would benefit from a more fulsome discussion of the role of systemic therapy in recurrent pilocytic astrocytoma (section 4.1) and of the risks with radiation therapy of transformation toward a more aggressive phenotype.

5. In section 4.3, the first paragraph should not be italicized.

Author Response

Reviewer #2

Point #2.1. The focus on methylation in the Introduction feels misplaced. If the authors wish to begin the review with a comment on methylation profiling of these rare tumours, I would suggest placing this discussion in a free-standing section after the introduction.

Response #2.1. We discussed methylation in the section [Introduction] because it is one of the major change in the new WHO 2021 classification and in order to introduce the readers to the concept of methylation which is detailed further for some entities. We have added as recommended by reviewer #1, a short paragraph in the section [Introduction] to introduce this major change. “In contrast to previous versions, the new World Health Organization (WHO) classification of primary CNS tumours, published in 2021, is based on histological features and/or molecular characteristics, such as specific genetic alterations and DNA methylation classes. Indeed, some tumor types need methylation analysis for accurate diagnosis (e.g. HGAP, DGONC)”

Point #2.2. I would suggest that the figures be revised to include a radiographic example of each tumour reviewed in the text.

Response #2.2: As requested by the reviewers #1 and #2, additional images have been added to the manuscript.

Point #2.3: In section 2.1, a brief description of Cowden disease would be valuable.

Response #2.3. We added a paragraph describing Cowden disease. “Cowden disease is a rare autosomic dominant condition, involving the PTEN gene. Patients present multiple and diffuse benign le-sions called hamartomas, mainly on the skin, breast, thyroid, as well as an increased risk to develop certain malignant lesions (breast, thyroid and digestive track). [7]”

Point #2.4. In section 3.4, I would ask the authors to speak to duration of mTOR inhibitor therapy and the concern of tumour recurrence with cessation of therapy.

Response #2.4. As suggested by the reviewer #2, we have expanded the paragraph and we have detailed mTORi therapy. “Surgery is preferred in case of hydrocephalus or signs of elevated intracranial pressure. mTOR inhibitors (mTORi) (e.g. everolimus) have become the first-line treatment for the management of SEGAs not requiring immediate surgical treatment, with studies showing a strong volume contraction as a result of the treatment and subsequent hydrocephalus prevention, as well as seizure reduction and even improvement of other manifestations of TSC. [97] [98] Due to risk of tumour regrowth in case of treatment discontinuation [99], a maintenance therapy mTORi may be needed. Overall incidence of AEs with mTORi is 30-74% (bronchitis, stomatitis, pyrexia), and 7% of patients discontinued the treatment. [99] [100] The EMINENTS study showed that a low-dose Everolimus maintenance therapy is as effective with less AEs than the standard dosage [101]. mTORi discontinuation has been associated to seizure relapse [102].”

Point #2.5. The manuscript would benefit from a more fulsome discussion of the role of systemic therapy in recurrent pilocytic astrocytoma (section 4.1) and of the risks with radiation therapy of transformation toward a more aggressive phenotype.

Response #2.5. As suggested by the reviewer, we have discussed more broadly different treatment options in recurrent PA. “The first-line treatment is surgery, preferably complete surgical resection, followed by observation [108] [104]. GTR reduces the risk of recurrence compared to STR (27% vs 73%) [109]. No study has confirmed a clear benefit of adjuvant chemotherapy/radiotherapy after a surgical resection, with two major studies finding a negative role of RT in the management of PAs concerning OS, but all RT regimens were considered all together [110] [111]. Radioinduced tumours have been described several years after pediatric PAs irradiation [112].

Stereotactic Radiotherapy/radiosurgery (SRT/SRS) could be proposed in adult patients and in case of midline/brainstem lesions: SRT was shown to be effective in controlling residual PAs without serious side-effects [113], and effectively improving PFS but not OS in adult patients [114]. A multicentric retrospective study showed favorable long term PFS and OS in patients with PAs after treatment with Gamma Knife SRS, either in first line setting or as a salvage treatment[115], and a high rate of pseudoprogression cases within 12 months [116]. A review concluded SRS as a safe and promising therapeutic in PA management [117], and should be discussed case by case.

Chemotherapy (CT) is a valid choice in optic gliomas or in pediatric populations with local recurrence [118]. Leptomeningeal dissemination is very rare [119] but exists and can mimic a DLGNT at initial diagnosis. Bevacizumab was shown to induce a durable response in recurrent PA [120] [121], Temozolomide can also be discussed [122].

MEK1/2 inhibitors target aberrant over-activation of the MAPK pathway: Selumetinib treatment showed responses and prolonged disease stability in patients with WHO grade 1 PA with either a KIAA1549-BRAF fusion or the BRAFV600E mutation [123] [124], as well as Trametinib in infants in case of CT failure [125] [126]. Dabrafenib is also efficient in controlling BRAFV600E PAs [127]. A case report showed a 19 month response in an adult patient with recurrent PA after Pemigatinib (pan-FGFR1) treatment [128].”

Round 2

Reviewer 2 Report

I thank the authors for their modified manuscript. 

Author Response

Thank you.